# Cross-Talks between RKIP and YY1 through a Multilevel Bioinformatics Pan-Cancer Analysis

**DOI:** 10.3390/cancers15204932

**Published:** 2023-10-11

**Authors:** Stavroula Baritaki, Apostolos Zaravinos

**Affiliations:** 1Laboratory of Experimental Oncology, Division of Surgery, School of Medicine, University of Crete, 71003 Heraklion, Greece; baritaks@uoc.gr; 2Department of Life Sciences, School of Sciences, European University Cyprus, 2404 Nicosia, Cyprus; 3Cancer Genetics, Genomics and Systems Biology Group, Basic and Translational Cancer Research Center (BTCRC), 1516 Nicosia, Cyprus

**Keywords:** pan-cancer analysis, RKIP, YY1, gene expression, mutations, immune infiltration, drug resistance

## Abstract

**Simple Summary:**

To better understand the potential implications of the cross-talk between RKIP and YY1 in cancer, we comprehensively analyzed different cancer types using bioinformatics approaches. Our study focused on analyzing the expression, mutation, immune infiltration, and drug resistance profiles of the two genes, using large-scale datasets from public repositories such as the Cancer Genome Atlas (TCGA) and the Cancer Proteome Atlas (TCPA). By integrating multiple data sources and applying advanced computational methods, we aimed to identify novel insights into the role of PEBP1/RKIP and YY1 in cancer, as well as potential therapeutic targets and biomarkers. Overall, our study provides new evidence that the cross-talk between RKIP and YY1 might be an important regulator of cancer progression and drug resistance, with potential clinical implications. By elucidating the complex interplay between these two genes, our findings may help in the development of more effective diagnostic and therapeutic strategies for cancer.

**Abstract:**

Recent studies suggest that PEBP1 (also known as RKIP) and YY1, despite having distinct molecular functions, may interact and mutually influence each other’s activity. They exhibit reciprocal control over each other’s expression through regulatory loops, prompting the hypothesis that their interplay could be pivotal in cancer advancement and resistance to drugs. To delve into this interplay’s functional characteristics, we conducted a comprehensive analysis using bioinformatics tools across a range of cancers. Our results confirm the association between elevated YY1 mRNA levels and varying survival outcomes in diverse tumors. Furthermore, we observed differing degrees of inhibitory or activating effects of these two genes in apoptosis, cell cycle, DNA damage, and other cancer pathways, along with correlations between their mRNA expression and immune infiltration. Additionally, YY1/PEBP1 expression and methylation displayed connections with genomic alterations across different cancer types. Notably, we uncovered links between the two genes and different indicators of immunosuppression, such as immune checkpoint blockade response and T-cell dysfunction/exclusion levels, across different patient groups. Overall, our findings underscore the significant role of the interplay between YY1 and PEBP1 in cancer progression, influencing genomic changes, tumor immunity, or the tumor microenvironment. Additionally, these two gene products appear to impact the sensitivity of anticancer drugs, opening new avenues for cancer therapy.

## 1. Introduction

The Raf kinase inhibitor protein (RKIP), also known as phosphatidylethanolamine-binding protein 1 (PEBP1), is a member of the phospholipid-binding protein family that regulates several central signaling pathways, including the Raf/MEK/ERK, G protein-coupled receptors (GPCRs), and NF-κB pathways. RKIP is a negative regulator of these pathways and has been shown to inhibit cell proliferation, invasion, and metastasis in multiple cancer types. In addition, RKIP plays a crucial role in drug resistance and apoptosis [1,2,3]. On the other hand, Yin Yang 1 (YY1) is a transcription factor that regulates a variety of biological processes, including cell proliferation, differentiation, and apoptosis. YY1 can activate or repress the expression of a wide range of genes by binding to specific DNA sequences [4,5]. YY1 targeting therapy could have clinical implications and potentially be used to improve patient outcomes [6]. Recently, YY1 was shown to repress RKIP in lung cancer [7], suggesting a potential link between these two genes.

Although RKIP and YY1 have different molecular functions, recent studies suggest that they might interact and modulate each other’s activity. Both gene products can modulate each other’s expression in an inverse relationship, through several regulatory loops [8]. For example, a recent study demonstrated that YY1 negatively regulates RKIP expression in lung cancer [9]. In addition, YY1 has been shown to interact either with transcriptional co-activators/co-repressors or with chromatin modulating enzymes to indirectly regulate the transcription of their targets [10]. Together, these findings suggest that a cross-talk between RKIP and YY1 might play a crucial role in cancer progression and drug resistance.

To better understand the potential implications of the cross-talk between the two genes in cancer, we conducted a comprehensive bioinformatics analysis across multiple cancer types. Our study focused on analyzing the expression, mutation, immune infiltration, and drug resistance profiles of RKIP and YY1, using large-scale datasets from public repositories such as the Cancer Genome Atlas (TCGA) and the Cancer Proteome Atlas (TCPA). By integrating multiple data sources and applying advanced computational methods, we aimed to identify novel insights into the role of RKIP and YY1 in cancer, as well as potential therapeutic targets and biomarkers.

Overall, our study provides new evidence that the cross-talk between RKIP and YY1 might be an important regulator of cancer progression and drug resistance, with potential clinical implications. Our findings shed light on the intricate interaction between these two genes, and have the potential to lay the foundation for the creation of improved diagnostic and therapeutic approaches for cancer.

## 2. Materials and Methods

### 2.1. Data Extraction

The TCGA mRNA expression, copy number variations (CNVs), and methylation data were retrieved from UCSC Xena (http://xena.ucsc.edu/ (accessed on 15 June 2023). The mRNA data (RSEM) were normalized to remove batch effects. The TCGA single nucleotide variants (SNV) data were obtained from the Synapse project (syn7824274; https://www.synapse.org/#!Synapse:syn7824274 (accessed on 15 June 2023). Calculation of the methylation levels of the two genes was based on their beta values.

#### 2.1.1. Differential Expression

Based on the normalized and batch corrected RSEM mRNA expression, we examined the differential expression of YY1 and PEBP1 across 14 TCGA cancer types and their paired normal tissues (thyroid cancer, THCA (n = 59); kidney renal papillary cell carcinoma, KIRP (n = 32); muscle-invasive bladder cancer, BLCA (n = 19); liver hepatocellular carcinoma, LIHC (n = 50); head and neck squamous cell carcinoma, HNSC (n = 43); breast invasive carcinoma, BRCA (n = 114); lung adenocarcinoma, LUAD (n = 58); prostate adenocarcinoma, PRAD (n = 52); esophageal carcinoma, ESCA (n = 11); kidney chromophobe, KICH (n = 25); lung squamous cell carcinoma, LUSC (n = 51); kidney renal clear cell carcinoma, KIRC (n = 72); stomach adenocarcinoma, STAD (n = 32); and colon adenocarcinoma, COAD (n = 26)). Differential expression analysis was based on the normalized and batch-corrected RSEM mRNA expression data. The fold change was calculated by mean(Tumor)/mean(Normal), and the *p*-values were estimated using *t*-test and adjusted by FDR. FDR values ≤0.05 were considered statistically significant. The differential gene expression analysis results were provided as bubble plots and boxplots.

#### 2.1.2. Expression and Survival

For expression and survival analysis, we explored 33 cancer types from GSCA [11]. Samples having a competing risk for death from cancer were filtered out (for DSS and DFI data). Tumor samples were divided into high and low YY1 (or PEBP1) expression groups, using the median value. We used the following prognostic indicators: overall survival (OS), progression-free survival (PFS), disease-specific survival (DSS), and disease-free survival (DFI). The survival status was fitted within the two groups using the R package *survival*.

We also explored patient survival using GEPIA2 [12]. The Cox proportional-hazards model and log rank tests were performed for each gene in every cancer. Data were presented as bubble plots and Kaplan–Meier curves for high and low expression of YY1 and/or PEBP1 in specific cancers to measure significant differences between curves. The genes with *p*-values < 0.05 from the Kaplan–Meier log rank test were statistically significant.

In addition, we used PrognoScan [13] as an additional tool for survival analysis, using various Gene Expression Omnibus (GEO) datasets.

#### 2.1.3. Expression and Molecular Subtype

We explored changes of gene expression in different molecular subtypes of HNSC, LUSC, COAD, STAD, LUAD, glioblastoma multiforme (GBM), BRCA, KIRC, and BLCA. Group comparison was performed using Wilcoxon and ANOVA tests.

#### 2.1.4. Expression and Stage

For expression and stage analysis, we investigated 4 stage types (pathologic, clinical, Masaoka (for thymoma, THYM), and the International Germ Cell Cancer Collaborative Group (IGCCCG) stage (for tenosynovial giant cell tumors, TGCT, only)) and the data of 9478 tumor samples from 27 cancer types (adrenocortical carcinoma, ACC; BLCA; BRCA; cervical squamous cell carcinoma and endocervical adenocarcinoma, CESC; cholangiocarcinoma, CHOL; COAD; diffuse large B-cell lymphoma, DLBC; ESCA; HNSC; KICH; KIRC; KIRP; LIHC; LUAD; LUSC; mesothelioma, MESO; ovarian cancer, OV; pancreatic ductal adenocarcinoma, PAAD; rectum adenocarcinoma, READ; skin cutaneous melanoma, SKCM; STAD; TGCT; THCA; THYM; uterine corpus endometrial carcinoma, UCEC; uterine carcinosarcoma, UCS; and uveal melanoma, UVM).

The pathologic, clinical, and Masaoka stages classify samples into stage I, II, III, and IV, while the IGCCCG classifies samples into good (n = 32), intermediate (n = 9), and poor (n = 2). Trend analysis was performed using the Mann–Kendall trend test.

#### 2.1.5. Expression and Pathway Activity

We estimated the difference in YY1 and PEBP1 gene expression between pathway activity groups (activation or inhibition), as defined by their median pathway scores. Reverse phase protein array (RPPA) data from TCPA (https://www.tcpaportal.org/tcpa/ (accessed on 15 June 2023) were used to assess the pathway activity score of 10 cancer-related pathways for 7876 samples across 32 cancer types.

RPPA data were median-centered and normalized by the standard deviation across all samples for each component to obtain the relative protein level. The pathway score was calculated as previously described [14].

The difference in pathway activity scores (PAS) between high and low gene expression groups was defined using the Student’s *t*-test. The *p*-values were adjusted by FDR, and the threshold of significance [15] was set at 0.05, as previously described in Ye et al. [16].

If the samples exhibiting elevated gene expression demonstrated significantly increased pathway activity (FDR ≤ 0.05), this suggested a potential activating impact on the pathway’s activity for the gene; conversely, if not, it indicated a potential suppressive influence on the pathway activity.

#### 2.1.6. Immune Infiltration and mRNA Expression

For immune infiltration analyses, we extracted a total of 10,995 samples from 33 cancer types in the TCGA, including 24 immune cells (18 T-cell subtypes, B cells, NK cells, monocytes, macrophages, neutrophils, and dendritic cells (DC)). The estimation of the abundance of the immune cells was based on specific gene set signatures (Appendix A), using the Immune Cell Abundance Identifier (ImmuCellAI) tool (http://bioinfo.life.hust.edu.cn/ImmuCellAI/#!/ (accessed on 15 June 2023) [17]. We then estimated the association between YY1 and PEBP1 mRNA expression and the immune cells’ infiltrates, using Spearman correlation analysis.

In addition, we analyzed associations between YY1 and PEBP1 mRNA expression and immune signatures/tumor immune cell infiltration in the TCGA database, using the R package “*UCSCXenaShiny*” [18]. In addition, we examined the Spearman correlations between molecular profiling and the tumor mutational burden (TMB), stemness, and microsatellite instability (MSI).

#### 2.1.7. Immune Infiltration and Mutations

We then estimated the difference in immune cells’ infiltrates between mutated genes (SNV) and wild-type (WT) groups through the Wilcoxon test. We also explored the association between YY1 or PEBP1 CNVs and immune cells’ infiltrates through Spearman correlation analysis. The infiltrates of 24 immune cells were evaluated through ImmuCellAI [15,17].

#### 2.1.8. Immune Infiltration and Methylation

We further examined the association between YY1 and PEBP1 methylation and immune cells’ infiltrates through Spearman correlation analysis. The *p*-values were FDR-adjusted.

#### 2.1.9. YY1 and PEBP1 Mutations

We also explored YY1 and PEBP1 SNVs in 10,234 samples across 33 cancer types, focusing on the presence of deleterious mutations, including missense, nonsense, frame-shift or in-frame insertions/deletions, and splice site mutations. We assessed differences in survival between mutant and WT groups, using the R package *survival*, the Cox proportional-hazards model, and log rank tests. Co-mutation was associated with the cancer clinical outcome, as previously described [19].

In addition, we downloaded CNV data of 11,495 TCGA samples and analyzed the genomic regions with significant amplifications or deletions, using GISTIC2.0 [20], The GISTIC scores (−2, −1, 1, 2) reflecting the CNVs in YY1 and PEBP1 in the selected cancer types were summarized and the results were depicted as oncoplots, pie plots, and bubble plots representing either heterozygous or homozygous amplifications and deletions of each gene per cancer.

#### 2.1.10. CNV and mRNA Expression

We correlated YY1 and PEBP1 CNVs with their corresponding mRNA expression levels, using Spearman’s correlation as previously described [21].

#### 2.1.11. CNV and Survival

The samples were divided into wild-type (WT), amplification (Amp.), and deletion (Del.) groups, and differences in their survival were assessed using the R package *survival* and the log rank test.

#### 2.1.12. Differential Methylation

We then set to explore the differential methylation between tumor and normal sample groups, using Illumina Human Methylation 450K (level 3) data. Generally, there are multiple methylation sites in the region of one gene. Therefore, there are multiple tags that store the methylation level of each site. Before differential methylation analysis, we performed correlation analysis to filter the sites most negatively correlated with YY1 and PEBP1 expression into this analysis.

#### 2.1.13. Differential Methylation and Survival

Before survival analysis, we performed correlation analysis to filter the site most negatively correlated with gene expression. We downloaded clinical data from 33 TCGA cancer types and the study by Liu et al. [22]. Uncensored data were left out. Samples with a competing risk for death from cancer were filtered out (for DSS and DFI data). The median methylation level was used to divide tumor samples into high and low methylation groups and survival differences between them were assessed using *survival*.

#### 2.1.14. Differential Methylation and YY1/PEBP1 mRNA Expression

We used Spearman’s test to get the correlation between gene mRNA expression and methylation levels.

### 2.2. Drug Sensitivity and YY1/PEBP1 Expression

We collected the half maximal inhibitory concentration (IC50) of 265 small molecules in 860 cell lines and the corresponding mRNA gene expression from two databases: Genomics of Drug Sensitivity in Cancer (GDSC; Release 8.4 (July 2022)) [16,23,24] and Genomics of Therapeutics Response Portal (CTRP v2) [25,26,27]. GDSC contains 265 small molecules IC50, 860 cell lines, and 17,419 genes, whereas CTRP contains 481 small molecules IC50, 1001 cell lines, and 18,900 genes. We explored the correlation between YY1/PEBP1 mRNA expression and drug IC50 using Pearson’s correlation test. Meanwhile, we used CellMiner Cross Database (CDB) (https://discover.nci.nih.gov/cellminercdb/ (accessed on 15 June 2023) to explore associations between YY1 (or PEBP1) expression levels and drug sensitivity, across all major cancer cell line pharmacogenomic data sources from NCI-DTP NCI-60, Sanger GDSC, and Broad CCLE/CTRP.

We then used ROC plotter tool (http://www.rocplot.org/ (accessed on 15 June 2023) [28] to explore the relationship between YY1/PEBP1 mRNA expression and sensitivity in endocrine therapy, anti-HER2 therapy, or chemotherapy in breast cancer, using the following parameters: response: pathological complete response (n = 1775), and treatment: ((any) chemotherapy; (any) endocrine therapy; (any) anti-HER2 therapy). A total of 1632 samples (1100 non-responders and 532 responders) met the above conditions for chemotherapy, 217 for anti-HER2 therapy (122 non-responders and 95 responders), and 64 (19 non-responders and 45 responders) for endocrine therapy. The Mann–Whitney test was used to compare expression differences between non-responders and responders to each type of therapy.

We also used the Tumor Immune Dysfunction and Exclusion (TIDE) algorithm [29,30] (http://tide.dfci.harvard.edu/ (accessed on 15 June 2023) to find associations between PEBP1/YY1 mRNA expression and immunotherapy (ICB) outcomes.

## 3. Results

### 3.1. Differential Expression of YY1 and PEBP1 in Pan-Cancer

We initially investigated pan-cancer the expression of YY1 and PEBP1, and found higher YY1 mRNA levels in HNSC, lung cancer (both LUAD and LUSC), ESCA, BRCA, and BLCA, compared to their corresponding normal tissues. In contrast, PEBP1 mRNA levels were significantly lower across all kidney cancer subtypes (KICH, KIRC, and KIRP), as well as in LIHC, HNSC, THCA, LUAD, and LUSC tumors (Figure 1A,B). The reverse expression pattern that the two genes exhibited in HNSC and lung cancers was previously described [31]. We further found that YY1 mRNA expression was correlated with clinicopathological stages in KIRC, KIRP, LIHC, OV, and SKCM, whereas PEBP1 mRNA expression exhibited the reverse pattern in these tumors, as well as in CESC, PAAD, and THCA (Appendix A). Trend analysis also revealed differences of YY1 mRNA expression between the pathologic and clinical stages in KIRP (pathologic stage: *p* = 1.27 × 10^−4^, FDR = 3.23 × 10^−3^, and clinical stage: *p* = 2.25 × 10^−5^, FDR = 3.19 × 10^−3^), as well as differences of PEBP1 mRNA expression across the pathologic stages of LIHC (*p* = 3.81 × 10^−3^, FDR = 0.11) and THCA (*p* = 3.08 × 10^−3^, FDR = 0.01), among others (Figure 1C and Appendix A).

In addition, we found that YY1 mRNA expression correlates with the molecular subtypes in BLCA (Non-Papillary vs Papillary; *p* = 0.049), BRCA (Basal, Her2, Luminal A, Luminal B and Normal-like; *p* = 9.7011 × 10^−9^), GBM (Classical, G-CIMP, Mesenchymal, Neural and Proneural; *p* = 0.046), KIRC (*p* < 0.0001), LUAD (*p* = 0.0016), LUSC (*p* = 0.009), and STAD (*p* = 0.0012), whereas that of PEBP1 correlates with GBM (*p* = 0.013), KIRC (*p* = 2.76 × 10^−30^), LUAD (subtypes 1–6; *p* = 0.00005), and LUSC (basal, classical, primitive, and secretory; *p* = 0.0007) (Figure 1D and Appendix A).

### 3.2. Pan-Cancer Analysis of Correlations between YY1/PEBP1 mRNA Expression and Patient Survival

To discover the prognostic value of YY1 and PEBP1 in pan-cancer, we analyzed differences in patient survival between high and low expression groups of the two genes, and found that the expression of both genes significantly (*p* < 0.05) correlates with survival in different tumor types (Appendix A).

High YY1 mRNA levels correlated with worse survival in HNSCC (OS and DSS), KIRC (OS, DSS), and KIRP (OS, PFS, DSS, and DFI) among other tumors (Figure 1E and Appendix A; e.g., LUAD (DSS), PAAD (OS, PFS, DSS, and DFI), PCPG (OS and PFS), PCPG (DFI), SARC (DFI), STAD (DSS), and THCA (PFS and DFI)). In READ and THYM on the other hand, high YY1 mRNA levels correlated with better DFI and DSS, respectively.

Similarly, high PEBP1 mRNA levels correlated with better survival in KICH (PFS and DFI) (Figure 1E and Appendix A), among other tumors (e.g., KIRP (OS and DSS), LIHC (OS, PFS, and DSS), LUAD (OS, PFA, DSS, and DFI), MESO (OS and PFS), PAAD (OS, PFS, and DSS), and UVM (OS, PFS, and DSS)).

We also used GEPIA2 survival analysis to explore the (overall and disease-free) survival maps of the two genes in pan-cancer. Our findings indicate that in ACC, PCPG, PAAD, and PRAD the overall survival of the high-YY1 expression group was significantly (*p* < 0.05) higher than that of the low expression group. In addition, in ACC, MESO, and UVM, the disease-free survival of the high-YY1 expression group was significantly higher than that of the low expression group. On the other hand, in KIRC, CESC, UCEC, and UVM, the overall survival of the high-PEBP1 expression group was significantly lower than that of the high expression group. Also, in DLBCL, high PEBP1 expression correlated with better DFS (Appendix A).

Using univariate Cox proportional-hazard regression (HR) model to predict the prognostic risk of the two genes in pan-cancer, we found that YY1 was an adverse prognostic factor (*p* < 0.05, HR > 0) for OS in PAAD, LUAD, and KIRP and a protective prognostic factor in OV and KIRC (*p* < 0.05, HR < 0). YY1 was also an adverse prognostic factor for PFI in UVM, LUAD, BLCA, and ACC and a protective prognostic factor in OV and KIRC. For DSS, YY1 was an adverse prognostic factor in PRAD, PAAD, LUAD, and KIRP and a protective prognostic factor in OV and KIRC, while for DFI, it was an adverse prognostic factor in PAAD, CESC, and ACC, and a protective prognostic factor in OV (Appendix A).

Likewise, PEBP1 was an adverse prognostic factor (*p* < 0.05, HR > 0) for OS in SKCM and a protective prognostic factor (*p* < 0.05, HR < 0) in PAAD, OV, LUAD, KIRP, and KIRC. PEBP1 was also an adverse prognostic factor for PFI in DLBCL and a protective prognostic factor in PAAD, OV, KIRP, and KIRC. For DSS, PEBP1 was a protective prognostic factor in PAAD, OV, KIRP, and KIRC, while for DFI, it was an adverse prognostic factor in PCPG, and a protective prognostic factor in THCA (Appendix A).

Overall, the analysis of multiple prognostic datasets pan-cancer revealed the cancers in which YY1 and PEBP1 significantly correlate with a good or bad prognosis, respectively.

We further used *PrognoScan* for survival analysis using various GEO datasets. We constructed a univariate Cox proportional-hazard regression model to predict the prognostic risk of YY1 and PEBP1 in pan-cancer, and found that the two genes associate with the prognoses of various tumors (Appendix A). For example, our analysis showed that YY1 is an adverse prognostic factor in AML, B-cell lymphoma, DLBCL, colorectal cancer, uveal melanoma, lung cancer, ovarian cancer, skin cancer, and liposarcoma (*p* < 0.05, HR > 0) and a protective prognostic factor in astrocytoma, breast cancer, gliomas, ovarian cancer, B-cell lymphoma, breast cancer, and renal cell carcinoma (*p* < 0.05, HR < 0).

On the other hand, PEBP1 was shown to act as an adverse prognostic factor in brain cancer (meningioma), colorectal cancer, breast cancer, ovarian cancer, and AML and a protective prognostic factor in bladder cancer, lung cancer, adenocarcinoma, and liposarcoma (Appendix A).

Taken together, we used multiple prognostic datasets for pan-cancer and preliminarily revealed that the high mRNA levels of YY1 and PEBP1 significantly correlate with a poor prognosis for different cancer patients.

### 3.3. Pathway Activity in Pan-Cancer

We then set to explore the differences of the activity of 10 pathways, between high and low YY1/PEBP1 mRNA expression. Our results reveal a strong inducing (activating) effect of YY1 mRNA expression on cell cycle (31%), apoptosis (16%), and DNA damage (16%) pathways, pan-cancer. YY1 mRNA expression also had an inhibitory effect on the EMT (12%), hormone ER (12%), RASMAPK (19%), and RTK (12%) pathways, pan-cancer.

PEBP1, on the other hand, had an activating effect on the hormone AR (19%), hormone ER (9%), PI3KAKT (9%), RTK (9%), and RASMAPK (6%) pathways, as well as a strong inhibitory effect on the apoptosis (22%) and EMT (25%), and a less strong effect on the cell cycle (9%), PI3KAKT (9%) and RTK (9%) pathways (Figure 2A and Appendix A).

For example, in bladder cancer, low PEBP1-expressing tumors had significantly higher activity scores in the apoptosis (FDR = 3 × 10^−6^) and EMT (FDR = 1.2 × 10^−7^) pathways, compared to high-expressing BLCA tumors. In contrast, high-expressing PEBP1 BLCAs had higher pathway activity scores in the DNA damage (FDR = 3.4 × 10^−7^) and hormone AR (FDR = 4 × 10^−20^) pathways (Figure 2B). BLCA tumors with high YY1 expression on the other hand, exhibited a reverse profile in the pathway activity scores of the cell cycle pathway and DNA damage pathway (higher PAS), as well as lower PAS in the EMT and hormone ER pathways, compared to BLCA tumors with lower YY1 expression (Figure 2B).

### 3.4. Correlation between YY1 and PEBP1 mRNA Expression and Immune Infiltration in Pan-Cancer

We next explored the correlation between YY1/PEBP1 mRNA levels and immune infiltration in pan-cancer. To this purpose, we evaluated the infiltrates of 24 immune cells through ImmuCellAI. Our findings highlight significant negative correlations (*p* < 0.001 and FDR < 0.01) between PEBP1 mRNA expression and infiltration score, as well as infiltration in B cells, central memory T cells, nTreg cells, cytotoxic T cells, dendritic cells (DC), macrophages, and iTregs in KICH. In contrast, PEBP1 mRNA expression was positively correlated with γδ T cells, NKT, neutrophils, Th17, and Th2 in KICH (Figure 3A,B and Appendix A). Likewise, YY1 mRNA levels correlated negatively with NK cells and positively with central memory T cells and nTreg cells in KICH (Figure 3A and Appendix A).

Collectively, our findings reveal that PEBP1 expression mainly associates with the infiltration of specific immune cells in KICH, among other tumors.

### 3.5. Correlation between YY1 (and PEBP1) Mutation Status and Immune Infiltration in Pan-Cancer

First, we explored the YY1 and PEBP1 gene mutation rate in pan-cancer. As expected, no significant differences were found due to the small mutation rate of both genes, in pan-cancer. Nevertheless, we found that 15% of uterine corpus endometrial carcinomas (UCEC) and 8% of skin melanomas (SKCM) harbored YY1 mutations. The mutation rate for PEBP1 was 6% in both tumor types (Appendix A). Overall, neither gene was mutated at high levels. Just 15 out of 531 (2.82%) of UCEC tumors were found to harbor YY1 mutations, and 6/531 (1.13%) of them had mutations in PEBP1 (Appendix A). These mutations affect both the activation, repression, and zinc finger domains of the YY1 gene locus. In a similar manner, the few PEBP1 mutations were scattered along the gene locus (Appendix A).

We then explored differences of immune cell infiltration between mutant and WT PEBP1 or YY1 tumors. We found a significant enrichment of Th2 cells in PEBP1 mutant skin melanomas (logFC = 0.304, *p* = 0.035), as well as an enrichment of B cells in YY1 WT skin melanomas (logFC = −0.56, *p* = 0.018) (Figure 3C and Appendix A). Th2 cells produce IL-4 and IL-10 and favor tumor growth by inhibiting the host immune system [32]. B cells, on the other hand, confer several anti-tumor roles through production of effective tumor-clearing antibodies, predominantly IgG1, mediating antibody dependent cell-mediated cytotoxicity (ADCC) and antibody dependent cell-mediated phagocytosis (ADCP), and facilitating complement activation [33].

In addition, we noted significant enrichment of iTreg (logFC = 0.41, *p* = 0.016) and Th1 cells (logFC = 0.66, *p* = 0.013) in YY1 mutant UCECs, as well as enrichment of mucosal-associated invariant T (MAIT) cells in WT YY1 UCECs (logFC = −0.408, *p* = 0.003) (Figure 3C and Appendix A).

iTreg cells are involved in an immunosuppressive tumor microenvironment (TME) and their targeting is a promising antitumor immunotherapy [34].

The function of Th1 cells is to activate macrophages and neutrophils; they are critical for host defense against intracellular pathogens such as *M. tuberculosis* [35].

MAIT cells are a newly described subset of T cells found in the blood and are enriched in many tissues, particularly in the liver. Human MAIT cells mainly express the CD8α coreceptor (CD8+), with a smaller subset lacking both CD4 and CD8 (double-negative, DN) [36].

Together, our results suggest that PEBP1 and YY1 mutations associate with the infiltration of specific immune cells in skin melanoma and uterine corpus endometrial carcinomas, respectively.

### 3.6. Correlation between YY1 (or PEBP1) CNV and Immune Infiltration in Pan-Cancer

We first assessed the percentage of YY1 and PEBP1 (heterozygous and homozygous) CNVs in pan-cancer. Our results reveal a large distribution of (mainly) heterozygous CNVs affecting both genes, across almost all tumor types. Of note, the highest percentage of CNVs was detected in UCS, OV, ACC, GCT, CHOL, and LUSC. CNVs affecting YY1 were almost double in number compared to those affecting PEBP1 (1.4% vs. 0.8%), whereas a significant number of tumors (mainly TGCT, ACC, KICH, HNSC, LUSC, BLCA, LUAD, SARC, ESCA, and UCS) was characterized by a high percentage of YY1 and PEBP1 amplifications and deep deletions (Appendix A).

To further explore the mechanisms underlying the abnormal mRNA expression of YY1 and PEBP1, we explored the relationship between each gene’s CNV and their mRNA expression levels in pan-cancer. Overall, we found various correlations between CNVs affecting both genes and immune infiltrates in BRCA, LUAD, KIRC, HNSC, and THCA, among others (Appendix A). For example, in BRCA, we spotted significant correlations between PEBP1 CNVs and CD8- naïve immune infiltrates (corr = 0.19, *p* = 1.86 × 10^−10^, FDR = 1.25 × 10^−9^), as well as macrophages (corr = 0.08, *p* = 7.89 × 10^−3^, FDR = 0.03) and neutrophils (*p* = 0.08, FDR = 0.03). PEBP1 CNVs were negatively correlated with infiltration score (corr = −0.09, *p* = 3.67 × 10^−3^, FDR = 0.01), NKT (corr = −0.13, *p* = 1.82 × 10^−5^, FDR = 1.27 × 10^−4^), B cell (corr. = −0.10, *p* = 1.14 × 10^−3^, FDR = 2.51 × 10^−3^), CD4 T cells (corr = −0.11, *p* = 4.69 × 10^−4^, FDR = 1.33 × 10^−3^), cytotoxic (corr = −0.09, *p* = 2.57 × 10^−3^, FDR = 7.34 × 10^−3^), exhausted T cells (corr = −0.08, *p* = 7.01 × 10^−3^, FDR = 0.02), NK (corr = −0.11, *p* = 4.16 × 10^−4^, FDR = 1.36 × 10^−3^), and Tfh infiltrates (corr = −0.12, *p* = 1.23 × 10^−4^, FDR = 5.14 × 10^−4^) in BRCA (Figure 3D and Appendix A).

Naïve CD8 T cells are found predominately in the circulation, spleen, and lymph nodes, where they survey the entire body for DCs presenting cognate antigens that will result in their activation. Their presence has been well described in breast cancer [37,38].

Macrophage infiltration associates with high vascular grade, reduced relapse-free survival, and decreased overall survival, and serves as an independent prognostic indicator of breast cancer [39,40]. In addition, high neutrophil infiltration has been associated with disease aggressiveness and therapy resistance [41].

The immune infiltration score is a method to quantify the immune cell infiltration within cancers to predict prognosis and chemotherapy effects in breast cancer [42]. Similarly, an infiltration for NKT cells has been reported in breast cancer [38,42,43], but this is the first mention of their association with CNVs in genomic regions affecting YY1 and PEBP1 to our knowledge.

Overall, these findings suggest that YY1 and PEBP1 CNVs might play a specific role in immune infiltration in breast cancer.

### 3.7. Correlation between YY1 (or PEBP1) Methylation and Immune Infiltration in Pan-Cancer

We then explored the methylation levels (beta values) of YY1 and PEBP1 across different TCGA tumors and found that PEBP1 is highly methylated in LGG, LAML, LUAD, LUSC, and HNSC, among other tumors, while YY1 is not significantly methylated in any tumor (Figure 4A,B).

Our next point of attention was to decipher associations between YY1/PEBP1 methylation (beta values) and immune cell infiltrates, across different tumor types. Our results show that methylation in both genes correlates with different immune cell infiltrates in COAD, DLBCL, LGG, LIHC, LUSC, PRAD, STAD, TGCT, THYM, and UVM (Appendix A). In addition, we found that PEBP1 methylation levels anticorrelate significantly with its mRNA expression in LUSC, SKCM, LGG, HNSCC, and LIHC (Figure 4C,D).

In THYM in specific, we found that PEBP1 methylation significantly correlates with infiltration of CD4 T cells, CD4 naïve T cells, CD8 T cells, CD8 naïve T cells, central memory, DC, effector memory, MAIT, macrophages, monocytes, neutrophils, Tfh, Th1, Th2, and nTreg cells in the tumor (Appendix A).

Additionally, YY1 methylation was found to correlate significantly with the infiltration score, as well as with infiltration of CD8 T cells, cytotoxic, effector memory, exhausted, gamma-delta (γδ), MAIT, NKT, neutrophils, Th1, Th17, Th2, and iTreg cells in UVM (Appendix A).

Taken together, our findings show that PEBP1 methylation might play a significant role in specific tumors, such as LUSC and HNSCC. They also suggest that mainly PEBP1 methylation (and YY1) might play a specific role in immune infiltration across different tumors, but especially in THYM and UVM.

### 3.8. Correlations between YY1/PEBP1 Expression and Immune Modulators, TMB, Stemness and MSI in Pan-Cancer

Stemness, TMB, and MSI in the TME are known to be related to antitumor immunity. We thus explored the associations between YY1 (or PEBP1) expression and immune signatures, tumor immune cell infiltration, TMB, stemness, and microsatellite instability (MSI) in the TCGA database, using the R package “*UCSCXenaShiny*”.

Figure 5A,B shows the correlations between YY1 (or PEBP1) expression and the infiltration levels of different immune cell subtypes. In specific, YY1 expression was positively correlated with CD4 naïve T cells in ACC; CD4 memory resting T cells across most of the tumors; M1 macrophages in BRCA, KIRC, KIRP, LIHC, LUAD, PRAD, and READ; M2 macrophages in BRCA, COAD, KIRC, LUAD, LUSC, and THCA; and activated DCs in BLCA, CESC, and UCEC; as well as negatively correlated with Tregs across most of the different tumors; follicular T helper cells in BRCA, HNSCC, KIRC, PCPG, PRAD, and TGCT; CD8 T cells in BRCA, CESC, GBM, HNSCC, SKCM, STAD, THCA, and UCEC; and memory B cells in BRCA, HNSC, KIRC, KIRP, LGG, LUAD, STAD and TGCT (Figure 5A).

Likewise, PEBP1 expression correlated positively with CD4 naïve T cells in BLCA, LIHC, and THCA; NK resting cells in LIHC and THCA; monocytes in ACC, KIRC, LGG, LIHC, and THCA; and resting mast cells in BRCA, HNSCC, KICH, KIRC, LGG, LIHC, LUAD, PRAD, SARC, and THCA; as well as negatively correlated with Tregs in ACC, KIRC, KIRP, LIHC, TGCT, and THCA; and activated CD4 memory T cells in BLCA, BRCA, KIRC, LUAD, LUSC, and THCA (Figure 5B).

We also investigated the correlations between YY1 (or PEBP1) expression and the abundance of infiltrating immune cells. Overall, YY1 expression correlated significantly with most of TIMER’s immune signatures across most of the different tumor types, including T cell C8+, neutrophil, DC, macrophage, and B cell (Figure 6A). In contrast, PEBP1 expression was generally anti-correlated with the abundance of infiltrating immune cells (Figure 6B).

Apart from the TME, stemness, TMB, and MSI are also related to antitumor immunity. We thus analyzed the relationships between YY1 (or PEBP1) expression and the stemness, MSI, and TMB across different tumors.

YY1 was positively correlated with stemness in ACC (r = 0.242, *p* = 0.035), BLCA (r = 0.28, *p* < 0.05), BRCA (r = 0.176, *p* < 0.05), ESCA (r = 0.298, *p* < 0.05), LAML (r = 0.256, *p* = 0.001), LUAD (r = 0.401, *p* < 0.05), LUSC (r = 0.456, *p* < 0.05), PCPG (r = 0.249, *p* = 0.001), PRAD (r = 0.166, *p* < 0.05), STAD (r = 0.357, *p* < 0.05), and TGCT (r = 0.325, *p* < 0.05), and negatively correlated with stemness in KICH (r = −0.385, *p* < 0.05) and KIRP (r = −0.112, *p* = 0.046) (Figure 7A).

Likewise, PEBP1 expression was positively correlated with stemness in ACC (r = 0.368, *p* = 0.001), BLCA (r = 0.115, *p* = 0.018), COAD (r = 0.167, *p* = 0.003), ESCA (r = 0.343, *p* < 0.05), LAML (r = 0.354, *p* < 0.001), OV (r = 0.162, *p* = 0.005), PCPG (r = 0.234, *p* = 0.001), PRAD (r = 0.177, *p* < 0.001), SKCM (r = 0.127, *p* = 0.006), STAD (r = 0.131, *p* = 0.006), TGCT (r = 0.243, *p* = 0.002), and THCA (r = 0.124, *p* = 0.003). PEBP1 was also negatively correlated with stemness in KICH (r = −0.539, *p* < 0.05), LUAD (r = −0.138, *p* = 0.001), PAAD (r = −0.477, *p* < 0.05), SARC (r = −0.171, *p* = 0.006), THYM (r = −0.636, *p* < 0.05), and UVM (r = −0.267, *p* = 0.017) (Figure 7B).

In addition, YY1 expression was positively correlated with MSI in BLCA (r = 0.092, *p* = 0.01), BRCA (r = 0.092, *p* = 0.004), CESC (r = 0.112, *p* = 0.049), GBM (r = 0.164, *p* = 0.038), LGG (r = 0.09, *p* = 0.04), LUAD (r = 0.17, *p* < 0.01), LUSC (r = 0.166, *p* = 0.001), PCPG (r = 0.157, *p* = 0.036), PRAD (r = 0.115, *p* = 0.007), READ (r = 0.28, *p* = 0.011), and SKCM (r = 0.282, *p* = 0.003) (Figure 7C).

The expression of PEBP1, on the other hand, was positively correlated with MSI in CESC (r = 0.196, *p* = 0.001), HNSC (r = 0.101, *p* = 0.019), LUSC (r = 0.129, *p* = 0.007), and PAAD (r = 0.206, *p* = 0.007) (Figure 7D).

Regarding TMB, YY1 expression was positively correlated in BRCA (r = 0.061, *p* = 0.047), COAD (r = 0.176, *p* = 0.004), DLBC (r = 0.31, *p* = 0.034), LUAD (r = 0.245, *p* < 0.05), and SKCM (r = 0.111, *p* = 0.016) (Figure 7E).

PEBP1 expression was further positively correlated with TMB in SKCM (r = 0.119, *p* = 0.01) and STAD (r = 0.132, *p* = 0.007), and negatively correlated with it in LUAD (r = −0.141, *p* = 0.001), OV (r = −0.122, *p* = 0.04), and THCA (r = −0.13, *p* = 0.004 (Figure 7F).

Altogether, the above findings indicate that YY1 and PEBP1 expression associates with the TME and anti-tumor immunity.

### 3.9. Correlation between YY1 (or PEBP1) Expression and Drug Sensitivity in Pan-Cancer

Finally, we collected the IC50 of various drugs across different cancer cell lines from the GDSC and CTRP databases and associated them with the corresponding YY1 and PEBP1 mRNA levels.

Collectively, we found several correlations. Interestingly, YY1 mRNA expression correlated positively with sensitivity in Dabrafenib, and negatively with sensitivity in QL-XII-61 and YM201636 (to a lower level) in CTRP. In addition, PEBP1 mRNA expression was shown to correlate positively with sensitivity in Afatinib, CCT018159, and Gefitinib, and negatively with many other drugs, including Dabrafenib, Nutlin-3a (-), EH 1864, MP470, SB52334, QL-XII-61, and YM201636 (Figure 8A).

Likewise, YY1 mRNA expression was found to correlate negatively with BI-2536, COL-3, CR-1-31B, GSK461364, LRRK2-IN-1, and NSC19630, among other drugs in CTRP. PEBP1 mRNA expression, on the other hand, correlated positively with sensitivity in dasatinib and negatively with sensitivity in nutlin-3, PL-DI, PRIMA-1, PX-12, belinostat, ciclopirox, and necrosulfonamide, among many other drugs (Figure 8B).

Subsequently, we used the ROC plotter tool to analyze the relationship between YY1 (and PEBP1) expression and sensitivity in endocrine therapy, anti-HER2 therapy, or chemosensitivity in breast cancer. The results indicated that the expression levels of both genes were significantly correlated with the response to chemotherapy in breast cancer (Mann–Whitney test, *p* < 0.05) (Figure 8C). In specific, YY1 was downregulated in the chemotherapy-sensitive group, while PEBP1 was upregulated in the responders to chemotherapy. In contrast, neither YY1 or PEBP1 expression correlated with sensitivity to the other two kinds of treatment (endocrine and anti-HER2 therapy).

Furthermore, we used the CellMinerCDB database to explore the association between YY1 (or PEBP1) expression and sensitivity in the aforementioned drugs from GDC and CTRP. We found that YY1 expression was significantly (*p* ≤ 0.05) negatively correlated with drug sensitivity of dabrafenib (r = −0.27, *p* = 0.05), whereas PEBP1 expression was positively correlated with sensitivity of the same drug (r = 0.43, *p* = 0.0014), as well as of nutlin-3a (r = 0.26, *p* = 0.05). PEBP1 was also negatively correlated with drug sensitivity of dasatinib (r = −0.32, *p* = 0.015) (Figure 8D). The above results suggest that YY1 and PEBP1 expression may serve as potential indicators for evaluating the sensitivity of certain chemotherapeutic drugs.

Finally, we explored the association between YY1/PEBP1 expression and patient response to immune checkpoint blockade (ICB), using TIDE. YY1 and PEBP1 were found to associate with several indicators of immunosuppression, including the ICB response outcome, T-cell dysfunction/exclusion levels, and phenotypes in CRISPR screens, across different patient cohorts, using regulator prioritization analysis (Figure 8E). High YY1 expression associated with a worse outcome of therapy with anti-CTLA-4 in skin melanoma (Van Allen, 2015 dataset) [44], but was related to better anti-PD-L1 therapy outcomes in metastatic urinary bladder cancer (Mariathasan, 2018 dataset) [45]. In addition, high PEBP1 expression was associated with a worse outcome of anti-PD-1 therapy in skin melanoma (Riaz, 2017 dataset) [46], but was related to better anti-D-L1 therapy outcomes in glioblastoma (Zhao, 2019 dataset) [47] (Figure 8F).

## 4. Discussion

Dysregulation of YY1 and PEBP1 expression patterns has been reported across different types of cancer. Nevertheless, most studies have focused on each gene per tumor. Here, we have systematically explored the patterns of gene expression, mutations, TME, immune infiltration, and drug sensitivity of the two genes, and we have evaluated their cross-talk in pan-cancer, for the first time.

YY1 is reported to be widely expressed in various tissues, acting either as an oncogene or as a tumor suppressor [48], thus playing a dual role in cancer development and progression in a cancer type-specific manner [49,50]. Accordingly, studies on dissecting the roles and clinical potential of YY1 in the TME revealed that although it is mostly overexpressed in the majority of cancers [51,52], different conclusions regarding the survival, prognosis, and other clinical implications of its expression analysis can be drawn across different tumors, as the functions of YY1 seem to be diversified in a pan-cancer context [52,53]. Similar conclusions have also emerged with respect to the diversity and frequency of the main YY1 phosphorylation patterns (S118 and S247) among different tumor types, as well as their associations with clinicopathological features [52]. On the other hand, downregulation or loss of RKIP expression has regularly been observed in many cancers, a fact that is mainly attributed to the altered epigenetic landscape (e.g., promoter methylation and epigenetic transcription silencing) rather than RKIP allele deletion or mutational events [54,55]. Although a pan-cancer computational analysis of RKIP expression and relevant associations with clinicopathological features including disease progression, metastasis, and therapeutic response are still missing from the literature, a general picture from individualized cancer types reveals that RKIP expression can be used as an independent prognostic marker for overall survival and disease-free survival [2,56]. Notably, the prognostic value of phosphorylated at Ser 153 RKIP protein levels (pSer153 RKIP), known to weaken its tumor suppressing activity, is still limited, as it has been studied in only a few cancer types [57,58,59,60,61].

Accordingly, the results of our analysis show that YY1 is considerably upregulated in HNSC, breast, bladder, and lung cancers (LUSC and LUAD), while PEBP1 is significantly downregulated in all kidney and lung cancer subtypes, as well as in LIHC, HNSC, and STAD. YY1 over-expression has also been previously reported ex vivo in different cancer types, including those of the breast [62], bladder [63], prostate [64], gastrointestinal [65], ovaries [66], esophagus [67], nervous system [68], pancreas [69], and skin [70]. YY1 mRNA levels were also recently explored in a systematic pan-cancer computational single analysis from Fu et al. [52]. In our study, we found similar findings, but we further addressed the interplay between the two genes (YY1 and RKIP). In addition, our findings on the distribution of RKIP mRNA levels across different cancer types corroborate its reported significant reduction in most malignancies, a fact known to support tumor growth and metastasis [71]. Accordingly, a computational analysis by Touboul et al. [60] revealed that PEBP1 is significantly under-expressed in most tumor types, including THCA, KIRP, KICH GBM, LUAD, CESC, and LUSC. In this study, we also report significant differences between high and low YY1 expression in different subtypes of BRCA, KIRC, LUAD, LUSC, and STAD, as well as between high and low PEBP1 expression in different subtypes of KIRC and lung cancers.

Our results further validate the association of high YY1 mRNA levels with different survival outcomes in different tumors. For example, we show that there is a significant association between increased YY1 expression and poor prognosis in HNSCC. Accordingly, Schnoell et al. [72] recently reported that high YY1 mRNA expression is indicative of worse overall survival in HNSCC patients and might be a suitable prognostic marker for their risk stratification, as well as a putative therapeutic target [73]. In addition, we show that there is a correlation between high PEBP1 expression and better prognosis in KICH. Although KICH is known to have the best prognosis among all kidney cancer types [74,75], reported findings from univariate and multivariate analyses of various cases of RCC revealed that reduced RKIP expression is an independent negative predictor for overall patient survival [76]. In this context, the urinary RKIP/pRKIP expression ratio has been further suggested to serve as a potential diagnostic and prognostic marker of the more aggressive clear cell renal cell carcinoma (ccRCC) [77]. Importantly, the reverse expression pattern of YY1 and RKIP that we observed in LUAD and LUSC might suggest an explicit YY1/RKIP cross-talk in lung cancers, with possible clinical significance. In support of this notion, Vivarelli et al. recently reported that YY1 and RKIP expression are inversely correlated in lung cancer, as the computational analysis of the deposited YY1-ChIP-Seq experiments revealed a negative regulation of RKIP by YY1, suggesting that both genes can be used for predictive diagnostic and prognostic purposes in this malignancy [9]. In this study, computational analysis revealed that YY1 negatively regulates RKIP expression in lung cancer, as corroborated by the deposited YY1-ChIP-Seq experiments and validated by their robust negative correlation.

In accordance, our preliminary findings from our studies investigating the direct role of YY1 in the transcriptional repression of RKIP in several solid malignancies (including LUSC) reveal a significant increase in RKIP promoter activity after YY1 silencing, thus supporting the inverse correlation of their expression levels, at least in lung malignancies (unpublished data).

We also explored the pathway activity scores of the two genes in pan-cancer. We found a strong inducing effect of YY1 expression in the cell cycle (31%), apoptosis (16%), and DNA damage (16%) pathways across different cancers, as well as a potent inhibitory effect on the RASMAPK (19%), EMT (12%), hormone ER (12%), and RTK (12%) pathways. PEBP1, on the other hand, could strongly induce the hormone androgen receptor (AR) (19%) pathway and inhibit the apoptosis (22%) and EMT (25%) pathways. These findings are in strong support of previous reports implicating YY1 in the regulation of a large number of genes being critical for cellular development, growth, differentiation, cell cycle, apoptosis, and DNA damage [64,78,79,80,81]. However, the role of YY1 in promoting or suppressing tumor growth and spread remains controversial and apparently conflicting, as different and still unclear molecular mechanisms of YY1 action may exist in a cancer-type specific context [82]. As such, and in support of its oncogenic function, YY1 has been shown to have proliferative and tumor promoting effects in several tumors, including gastric, breast, prostate, cervical, colon, neuroblastoma, and hepatocellular carcinomas [83,84], whereas in others, such as in pancreatic ductal adenocarcinoma, it can suppress the proliferation and migration of cancer cells [85]. Contrarily, we and others have identified a clear role of RKIP in the regulation of EMT and apoptosis in the tumor microenvironment, resulting in potent anti-metastatic and tumor suppressing effects in several tumor models [31,71,86]. Interestingly, among the underlying molecular signaling cascades currently suggested to mediate the RKIP activities in cancer cells, we have identified and proposed the dysregulation of the NF-κΒ/Snail/YY1/RKIP/PTEN circuitry, not only as an example of the RKIP/YY1 cross-talk in cancer cells, but also as a moderator and connector of several critical processes that take place in tumors, like autophagy, EMT, and resistance to apoptosis [87,88,89,90]. In addition, PEBP1 was recently identified as a newly-found target gene of androgens and AR [91,92]; however, the possible vice-versa regulation of AR pathway by RKIP, proposed in our analysis, is a novel finding that needs further investigation in terms of clarifying whether it concerns the pSer153 RKIP form, as it is in the case of pSer153 RKIP-mediated stimulation of the β-adrenergic receptor signaling [93]. Given that YY1 promotes the transcriptional activation of AR in prostate cells, which further cooperates with YY1 to activate expression of prostate specific antigen (PSA) and prostate stem cell antigen (PSCA), both known to contribute to prostate oncogenesis [94], a possible YY1/RKIP cross-talk in prostate cancer is also worth exploration [95].

Our study further detected several correlations between YY1 and PEBP1 mRNA expression and immune infiltration across different cancers. As expected, most of the associations for both genes followed a cancer type-specific pattern. Notably, across the majority of different tumors, positive correlations were established between YY1 and various immunosuppressing (iTreg and nTreg) and naïve/resting immune populations, including M2 macrophages and CD4+ naïve and memory resting T cells. In particular, in breast cancer models, it was reported that the intra-tumoral naïve CD4+ T cells and Tregs share overlapping TCR repertoires, suggesting that the existing Tregs in the tumor microenvironment may have originated by reprograming of tumor infiltrating naïve CD4+ T cells to Tregs, and therefore blocking the recruitment of naïve CD4+ T cells might reverse immunosuppression in this cancer type [96]. In contrast, in a considerable number of tumors, YY1 expression was negatively associated with intra-tumoral abundance of CD8 T cells, follicular T helper cells (Tfh), as well as with memory B cells, which are all known to promote the formation of an immunoreactive tumor microenvironment. Specifically, both Tfh and memory B cells advance the anti-tumor cellular and humoral immunities by promoting a chemokine-mediated CD8+ T cell and B-cell infiltration [97] and contributing, as APCs, to additional CD4+ T cell expansion, intratumorally [98]. Memory B cells have also been reported to possess cytotoxic functions against cancer cells by secreting IFN-γ, granzyme B, and TRAIL [99]. Corroborating our findings on the critical immunomodulatory role of YY1 in cancer progression, data derived by a recent pan-cancer analysis also demonstrate negative correlation between YY1 expression and infiltration of most immunoreactive cells, including CD8+ T cells, B cells, macrophages, dendritic cells, and neutrophils, in >10 different cancer tissues; while YY1 expression has been positively associated with the estimated infiltration value of cancer-associated fibroblasts [52]. Although the expression of YY1 in immune cells has been extensively studied, particularly under the prism of B and T cell development and exhaustion in cancer cells through the regulation of PD1 and LAG3 [100,101], it remains to be elucidated how YY1 expression interferes with the diverse immune cell infiltrates in the microenvironment of different tumors.

With respect to PEBP1, our analysis revealed that its expression is significantly negatively correlated with the infiltration of Tregs and memory CD4+ T cells in six cancer types, including lung tumors, while positively associated with the infiltration level of resting mast cells and NK cells in ten and two different malignancies, respectively, including different lung cancer subtypes. Li et al. found that aberrant activation of mast cells and CD4+ memory T cells was critical for cigarette smoking-induced immune dysfunction in the lungs, which further associated with tumor development and progression [102]. Accordingly, Zhang et al. suggested a prognostic model in HCC based on the negative correlation of the risk score with the content of the tumor-infiltrated resting mast cells [103]. The critical implication of RKIP in the modulation of the immune-composition of the tumor microenvironment by regulating the infiltration and activities of various immune cell populations has been reviewed by Gabriela-Freitas et al. [104]. Briefly, RKIP induction in triple negative breast tumor models was able to inhibit TAMs infiltration and their pro-metastatic activity in vivo and in vitro, through RKIP-mediated HMGA2 blockade, which eventually led to the reduction of numerous macrophage chemotactic factors, including CCL5 [105,106]. RKIP levels were further shown to be inversely correlated with the myeloid/lymphoid ratio, as well as with gene signatures associated with myeloid cell infiltration in melanoma tissues [107], while positively associated with signatures of effective T-cell responses in melanoma [107] and gastric cardiac adenocarcinomas [108]. Moreover, RKIP inhibition in CLL leads to elimination of CXCR4 expression by cancer cells [109], whereas RKIP was shown to play an important role in controlling mast cell-mediated allergic responses, specifically by negatively regulating mast cell activation [110]. The latter agrees with our observed positive association of RKIP with the infiltration of resting mast cells.

Overall, in the context of YY1/RKIP cross-talk in the regulation of tumor infiltration by immune cells, our results show that YY1 and PEBP1 expression scores were anti-correlated with infiltration levels of nTreg cells, B cells, neutrophils, and CD8 naïve T cells, as well as with the infiltration score and CD4+ T cells, Tfh, NK, MAIT, NKT, and Th2 cells. The immunomodulation within the tumor microenvironment undoubtfully plays crucial roles in tumor progression and metastasis, and the prognostic value of the suggested gene signatures seem to shed light on new and better disease management and therapeutic targeting. Nevertheless, further validation of the above anti-correlations remains to be elucidated.

Gene expression and cancer progression are regulated by genetic alterations in key genes. Here, we found various correlations between YY1/PEBP1 expression (or methylation) and genomic alterations, such as SNVs and CNVs that exhibited a cancer type-specific pattern. The genetic alteration status of YY1 has been previously reported in a pan-cancer context, and has demonstrated effects of YY1 mutation on gene expression in multiple myeloma and uterine cancer, as well as on YY1 expression in cervix cancer, head and neck cancer, lung squamous cell carcinoma, and melanoma [52]. In addition, the same study showed that only cholangiocarcinoma, DLBC, and KIRP cases with genetic alteration had copy number deletion of YY1, while the “amplification” type of CNVs was only seen in adrenocortical carcinoma and pheochromocytoma/paraganglioma tumors [52].

Furthermore, our analysis reveals a differential methylation pattern of both genes in pan-cancer. Importantly, we show that PEBP1 methylation is negatively correlated with its mRNA expression, in pan-cancer, with LUSC, SKCM, LGG, HNSC, and LIHC being the leading tumor entities in this anti-correlation. Similar studies are missing for RKIP in a pan-cancer context; therefore, our RKIP-associated findings are the first of this kind. Another interesting finding in our study is that the expression of the two genes is associated with TMB, stemness, MSI, and tumor immunity in different cancers. This is in accordance with individual findings on the associations of each gene with the aforementioned parameters [9,52,111,112,113]. Notably, RKIP expression associations with MSI have been studied only in colorectal cancer where they were identified as independent parameters [2].

Lastly, we found that both YY1 and PEBP1 gene expressions were also correlated with chemosensitivity and immunotherapy responses, in different types of tumors. Drug resistance is an important reason for the low five-year survival rate of cancer patients, and improving drug sensitivity is a significant topic in the field of cancer. Our findings corroborate previous reports by us and others, demonstrating the critical but opposite functions of YY1 and RKIP in the regulation of tumor sensitivities to drug and endogenous immune-mediated cytotoxic effects and/or immunotherapy [8,90,95,114].

We finally would like to report a limitation in our study, which is the lack of in vitro and/or in vivo validation experiments of our findings, which could further decipher the molecular function of the two genes in the setting of different tumor types.

## 5. Conclusions

In conclusion, our study unveiled the pivotal role of the cross-talk between YY1 and PEBP1 in tumor progression, encompassing genetic alterations, tumor immunity, and the tumor microenvironment, while also highlighting their potential influence on anticancer drug sensitivity, offering fresh insights and therapeutic targets for cancer treatment.

## Figures and Tables

**Figure 1 cancers-15-04932-f001:**
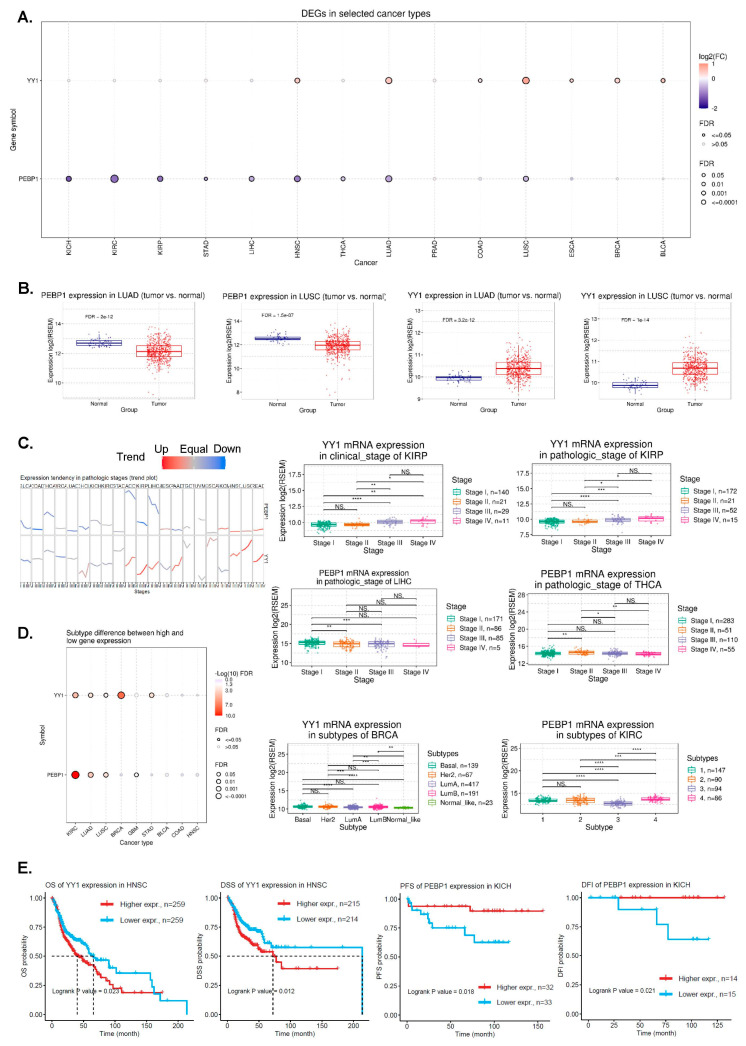
(**A**) The bubble plot presents the fold change of YY1 and PEBP1 across 14 cancer types, and the corresponding FDR through bubble color and size. The bubble color from purple to red represents the fold change between tumor vs normal. The size of the dots is positively correlated with the FDR significance. (**B**) PEBP1 and YY1 mRNA expression in lung cancers (LUAD and LUAD) compared to normal tissue. (**C**) The trend plot summarizes the trend of YY1 and PEBP1 mRNA expression from early stage to late stage in pan-cancer. The boxplots to the right are examples of differential YY1 and PEBP1 expression in KIRP, LIHC, and THCA. (**D**) The associations between different cancer subtypes and YY1/PEBP1 mRNA expression. The boxplots to the right are examples of differential YY1 and PEBP1 expression in molecular subtypes of BRCA and KIRC, respectively. (**E**) Kaplan–Meier plots depict examples of survival differences between high and low YY1 (and PEBP1) mRNA expression groups in HNSCC and KICH, respectively. * *p* < 0.05, ** *p* < 0.01, *** *p* < 0.001, **** *p* < 0.0001, NS, not significant.

**Figure 2 cancers-15-04932-f002:**
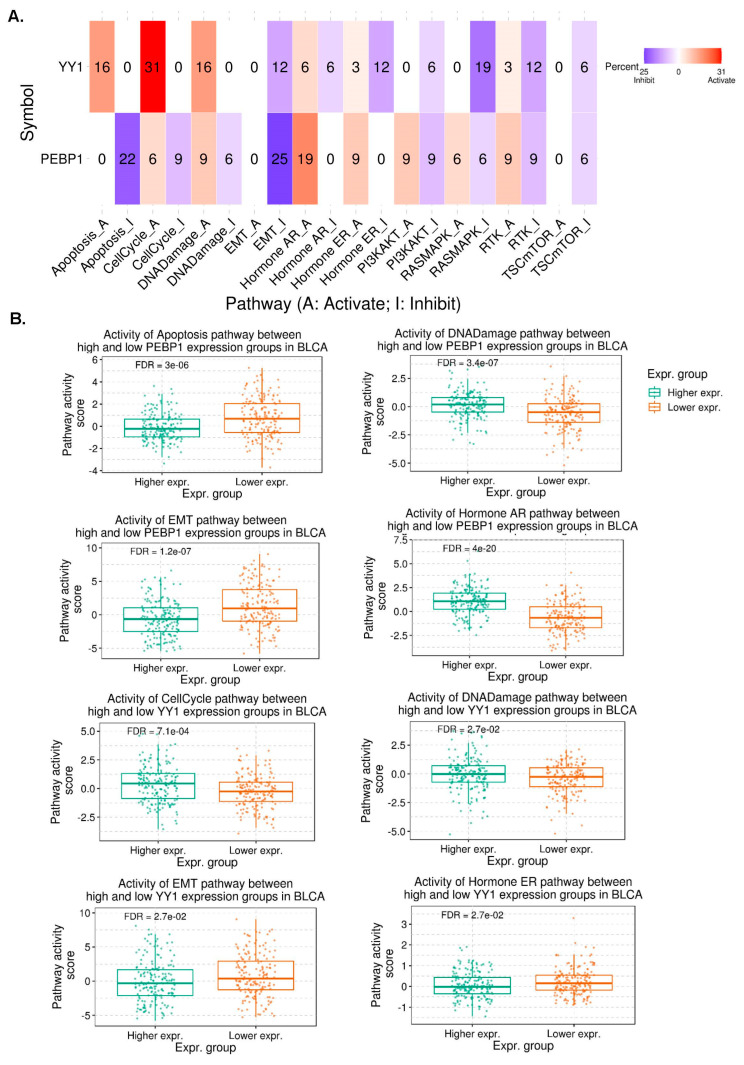
(**A**) The percentage (%) of cancers in which the mRNA expression of YY1 and PEBP1 has a potential effect on the activity of 10 cancer-related pathways (apoptosis, cell cycle, DNA damage, EMT, hormone AR, hormone ER, PI3KAKT, RASMAPK, RTK, and TSC-mTOR). The blue color means that this effect shifts towards inhibition of the pathway, while red means activation. The number in each cell indicates the percentage of cancer types in which each gene (YY1 and PEBP1) showed significant association (inducing or inhibitory) with a specific pathway, in pan-cancer. (**B**) Pathway activity scores (PAS) of high and low (YY1 and PEBP1) expression groups in BLCA tumors.

**Figure 3 cancers-15-04932-f003:**
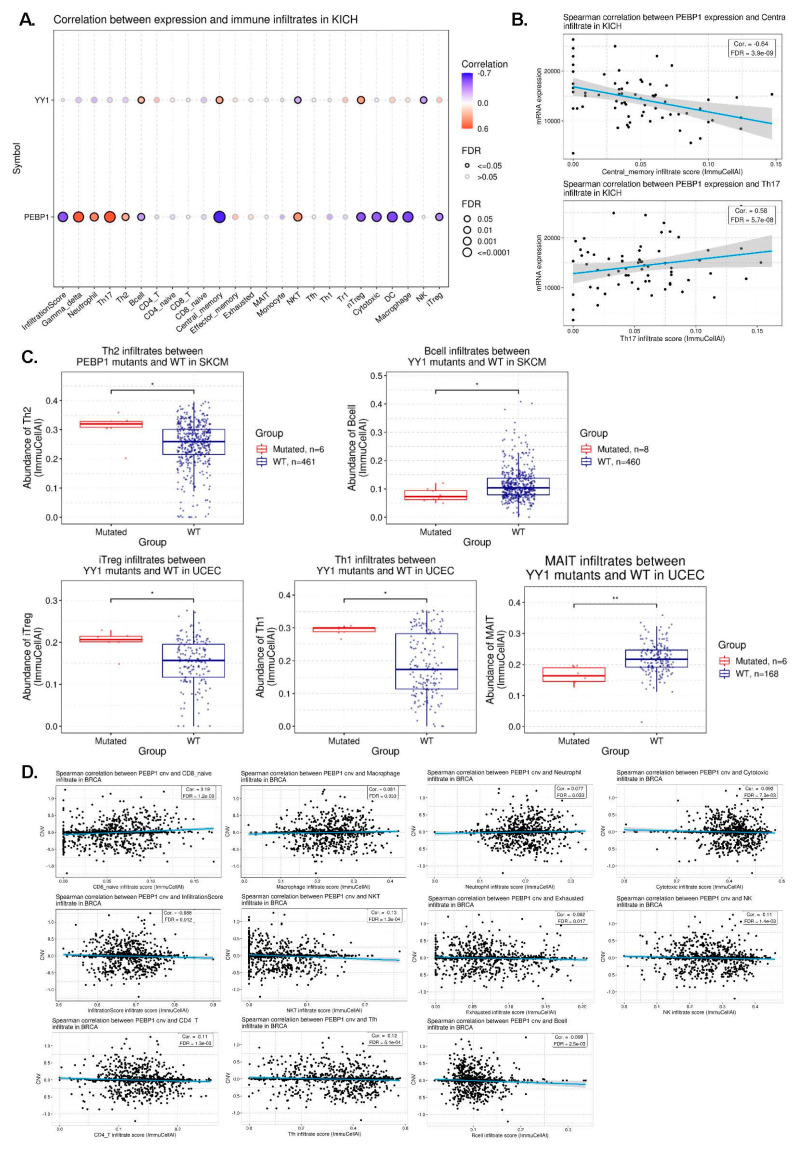
(**A**) Correlation between YY1 (and PEBP1) mRNA expression and immune infiltrates in KICH. (**B**) Spearman correlation between PEBP1, central infiltrate, and Th17 infiltrate in KICH. (**C**) Upper panel: Th2 infiltrates between PEBP1 mutants and WT, and B cell infiltrates between YY1 mutants and WT in SKCM. Lower panel: iTreg, Th1, and MAIT infiltrates between YY1 mutants and WT in UCEC. (**D**) Association between PEBP1 CNV and immune cell infiltrates in breast cancer, through Spearman correlation analysis. * *p* < 0.05, ** *p* < 0.01.

**Figure 4 cancers-15-04932-f004:**
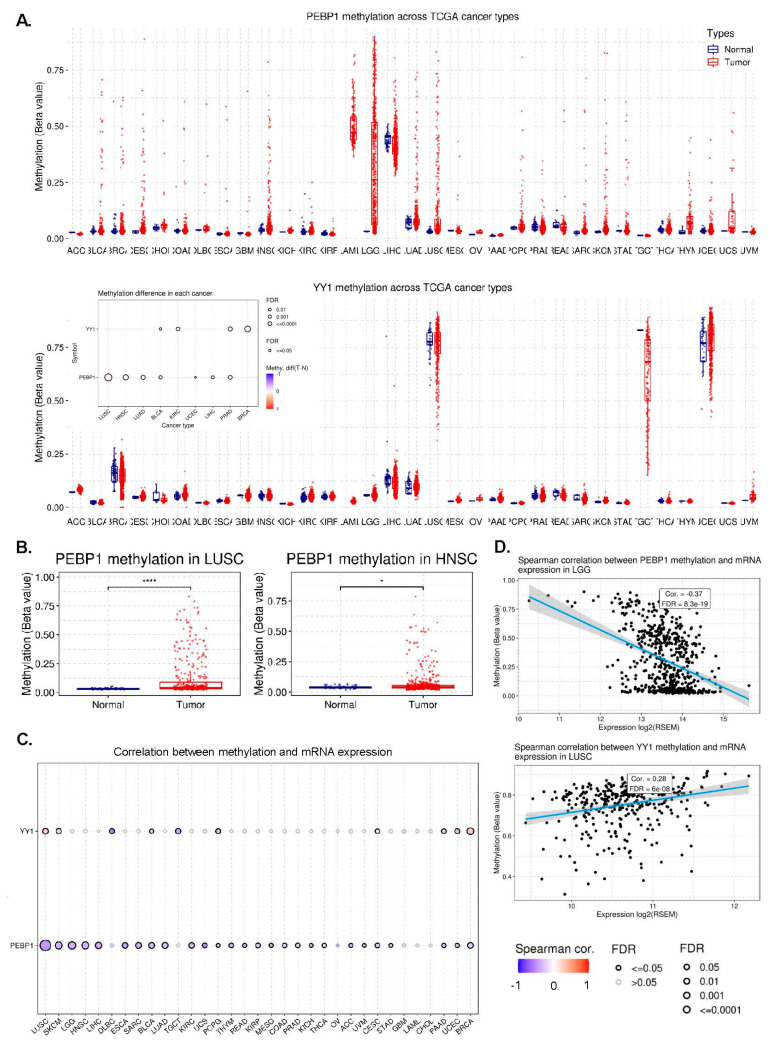
(**A**) Differential PEBP1 and YY1 methylation levels (beta values) between tumor and normal samples in pan-cancer. (**B**) Differential methylation levels of PEBP1 in LUSC and HNSCC. (**C**) Correlation between YY1 and PEBP1 methylation and their mRNA expression in pan-cancer. (**D**) Spearman correlations between PEBP1 (and YY1) methylation and their mRNA levels in LGG and LUSC, respectively. * *p* < 0.05, **** *p* < 0.0001.

**Figure 5 cancers-15-04932-f005:**
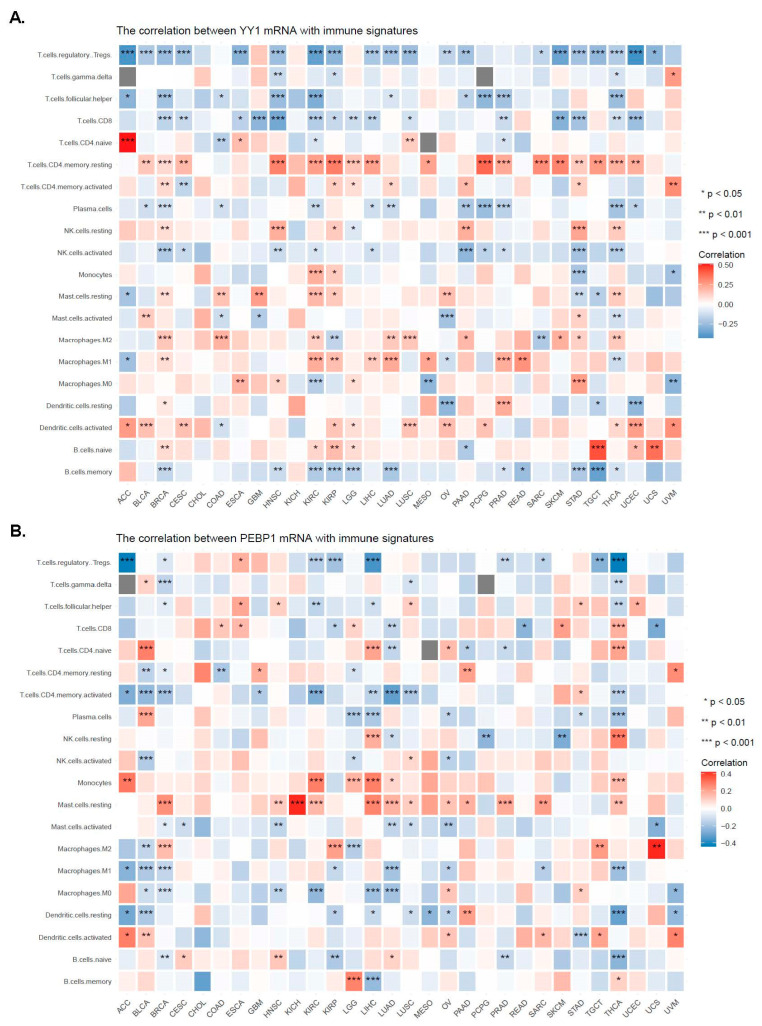
Correlations between the expression of YY1 (**A**) or PEBP1 (**B**) and the infiltration levels of different immune cell subtypes. * *p* < 0.05; ** *p* < 0.01; *** *p* < 0.001.

**Figure 6 cancers-15-04932-f006:**
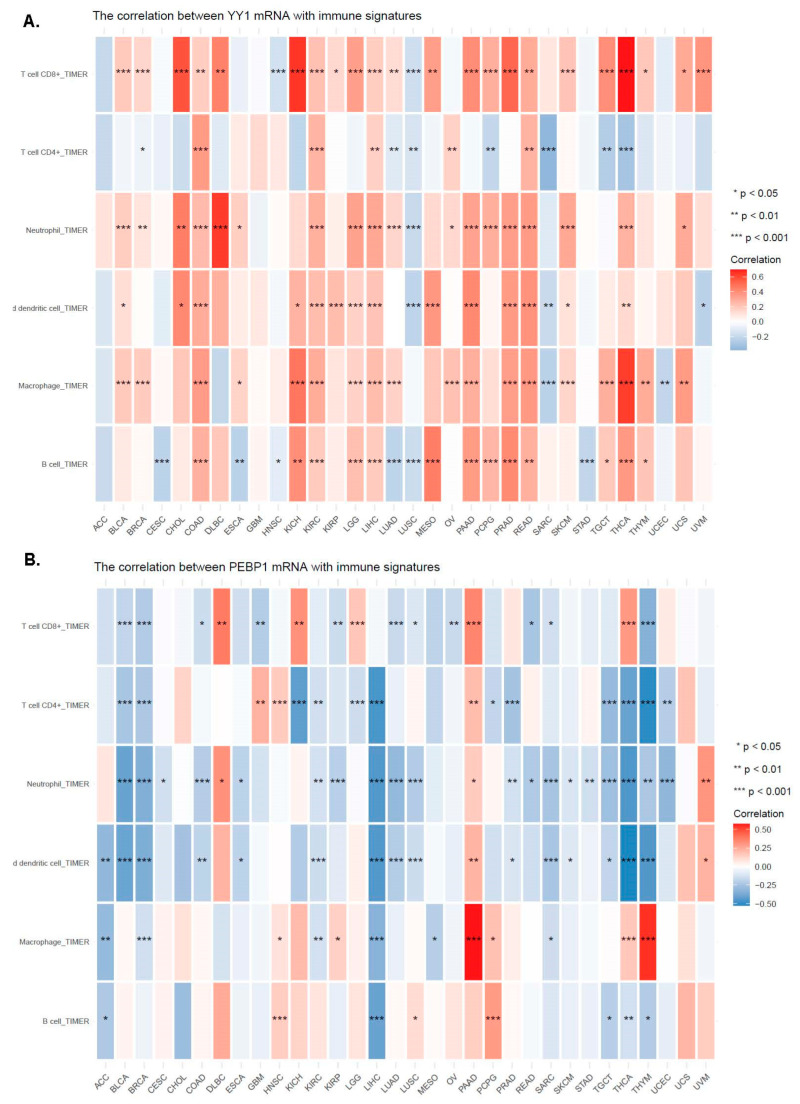
The expression of YY1 (**A**) correlated significantly with the following immune signatures from TIMER: T cell C8+, neutrophil, dendritic cell, macrophage, and B cell, pan-cancer. In contrast, PEBP1 expression was generally anti-correlated with the abundance of infiltrating immune cells (**B**). * *p* < 0.05; ** *p* < 0.01; *** *p* < 0.001.

**Figure 7 cancers-15-04932-f007:**
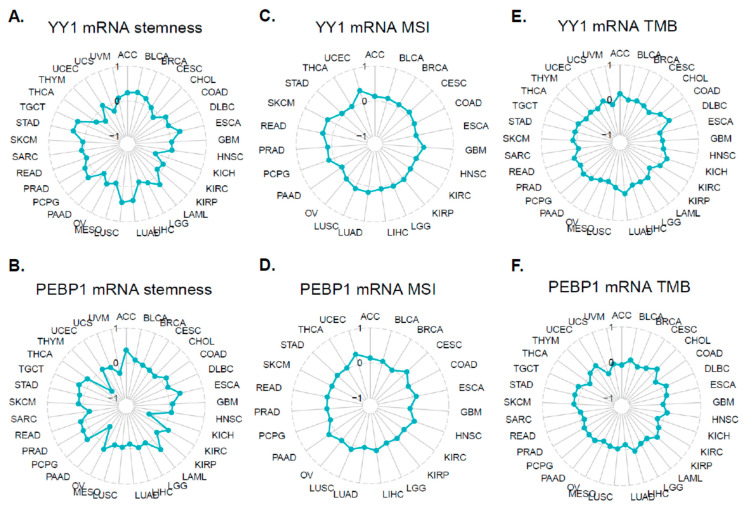
Relationships between the mRNA expression of YY1 (or PEBP1) and stemness (**A**,**D**)/MSI (**B**,**E**)/TMB (**C**,**F**) of the TCGA tumors.

**Figure 8 cancers-15-04932-f008:**
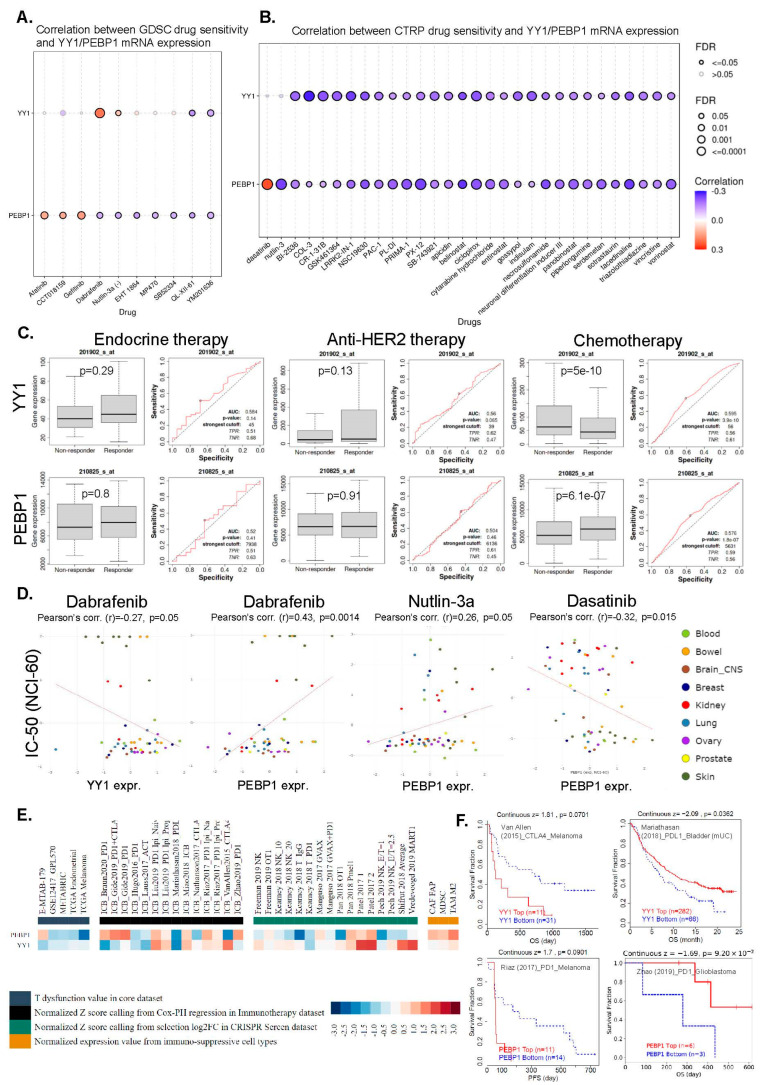
YY1 and PEBP1 expression correlates with chemosensitivity in pan-cancer. We analyzed the correlation of YY1 and PEBP1 mRNA levels with GDSC (**A**) and CTRP (**B**) drug sensitivity (IC50) in pan-cancer. The color from blue to red represents the correlation between each gene’s mRNA expression and IC50. The bubble size positively correlates with the FDR significance. A black outline border indicates an FDR < 0.05. (**C**) The ROC plotter tool was used to analyze the relationship between YY1 and PEBP1 expression and sensitivity in endocrine therapy; anti-HER2 therapy, and chemotherapy in breast cancer. *p* < 0.05 was considered statistically significant. (**D**) Drug sensitivity analysis of YY1 and PEBP. YY1 expression was negatively associated with the drug sensitivity of Dabrafenib across blood, bowel, brain, breast, kidney, lung, ovary, prostate, and skin tumors, whereas PEBP1 expression was positively associated with the drug sensitivity of dabrafenib and nutlin-3a, and negatively associated with the drug sensitivity of dasatinib, in the aforementioned tumor types. *p* < 0.05 was considered statistically significant. (**E**) The regulator prioritization clustering heatmap revealed the association of YY1 and PEBP1 with several indicators of immunosuppression. (**F**) Differences in OS (and PFS) of immunotherapy patients with high and low YY1 (or PEBP1) expression.

## Data Availability

All data supporting the reported results in this study were retrieved from the TCGA and GEO datasets.

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
