# Peer review of "Cross-Talks between RKIP and YY1 through a Multilevel Bioinformatics Pan-Cancer Analysis"

_cancers, 2023, doi:10.3390/cancers15204932_

Round 1

Reviewer 1 Report

Present article titled "Cross-talks between RKIP and YY1 through a multilevel bio-informatics pan-cancer analysis" by Baritaki et al is well researched. The authors have opted for an bioinformatic approach to analyze the online available datasets.

I have following comments:

The research design is not clear. Please add complete information about the datasets size and the analysis pipelines used.

How was batch effect corrected? Was is corrected on the basis of housekeeping gene expression across the samples ?

The authors have used PERB1 and RKIP interchangeably, which is creating a lot of confusion while reading. Please use only one gene symbol across the manuscript.

Which analysis predicted the crosstalk of RKIP/YY1? In my understanding the data shows that both of these genes have their individual impact on clinical characteristics of the disease. But how they are working together is not clear.

Figure2A is less informative and very complicated to understand. If possible please simplify.

English is fine.

Reviewer 2 Report

Baritaki and Zaravinos report in their study a pan-cancer analysis of RKIP/PEBP1 and YY1 mRNA expression and the relationship between these genes and clinical data. This is a well-done data mining analysis that shows intriguing relationships that could be further explored to develop diagnostic/therapeutic tools.

1. Authors must investigate if PEBP1 and YY1 mRNA expression is correlated in the different tumour types

2. Authors must clarify whether the different associations found between YY1 and PEBP1 mRNA and clinical data for a specific tumour type are, in fact, due to the mysregulation of the two genes (up or down) in the same tumours/patients ?

3. The same applies to CNVs. There is often a correlation between CNVs and gene expression. Authors must therefore present whether there is an association between YY1 and PEBP1 mRNAs and clinical/immune data as a consequence of the presence of CNV at YY1 and PEBP1 loci in these tumours.

4. Most of the work is based on data mining, and some validation, at least in cellulo, is needed and doable for certain findings (drug sensitivity…). This could validate that the associations identified are significant.

No comment.
